# Block-State Transformers

**Jonathan Pilault**[124*]   **Mahan Fathi**[123*]
**Orhan Firat**[1]   **Christopher Pal**[24]   **Pierre-Luc Bacon**[23]   **Ross Goroshin**[1]
[1]Google DeepMind   [2]Mila   [3]Université de Montréal   [4]Polytechnique Montréal

## Abstract

State space models (SSMs) have shown impressive results on tasks that require modeling long-range dependencies and efficiently scale to long sequences owing to their subquadratic runtime complexity. Originally designed for continuous signals, SSMs have shown superior performance on a plethora of tasks, in vision and audio; however, SSMs still lag Transformer performance in Language Modeling tasks. In this work, we propose a hybrid layer named Block-State Transformer (BST), that internally combines an SSM sublayer for long-range contextualization, and a Block Transformer sublayer for short-term representation of sequences. We study three different, and completely parallelizable, variants that integrate SSMs and block-wise attention. We show that our model outperforms similar Transformer-based architectures on language modeling perplexity and generalizes to longer sequences. In addition, the Block-State Transformer demonstrates more than *tenfold* increase in speed at the layer level compared to the Block-Recurrent Transformer when model parallelization is employed.

## 1   Introduction

Transformers have shown impressive performance on a wide range of natural language processing (NLP) tasks. While they have been primarily used for language modeling the Transformer architecture [40] has also been successfully applied to other tasks outside of the NLP and have mostly replaced Recurrent Neural Networks (RNNs). Several factors contribute to this success, including computational efficiency and architectural inductive biases that are well-suited for training on natural language tasks at scale. On the computational upside, Transformers are able to process tokens of a given input sequence in parallel, making the most of modern accelerator hardware. Moreover, the attention mechanism enables Transformers to find relationships in longer sequences by providing ready access to all the extracted information from past tokens when inferring the next token. Compared to RNNs and LSTMs [19], the benefits of self-attention are two-fold: (i) the capacity of what could be stored and directly accessible as context is drastically increased, and (ii) training on longer sequences is more stable [18, 23].

Given the remarkable achievements of Transformers in language modeling tasks, and their improved performance at scale on hard NLP tasks such as reasoning and question answering [2, 39, 6], the demand for deploying even deeper and larger networks is greater than ever before. An orthogonal scaling dimension, which could be potentially even more consequential, is the size of the input sequence. Despite the several advantages of Transformers over RNNs, it is still problematic to scale the input sequence length, again for both computational performance and quality reasons. Further, the Transformer's runtime is quadratic with respect to the input sequence length, which makes training these models increasingly expensive. Furthermore, Transformers with attention, that is local [8], sparse [4, 43, 36], low-rank approximated [41] or linearized via kernel methods [5, 22], notoriously struggle on long-input classification tasks [37]. Vanilla transformers can be unstable when trained

---

*Equal Contribution.

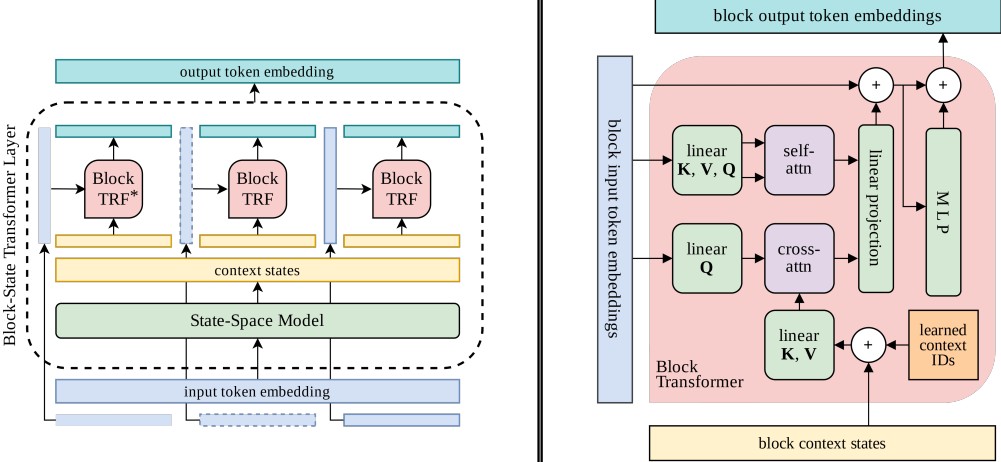

Figure 1: Block-State Transformer layer. The BST-SH layer is illustrated on the left, and includes a state space model (SSM, in green) and Block Transformers (in red). For demonstration purposes the sequence is divided into 3 blocks in the picture. The details of the Block Transformer sublayer are on the right. *TRF = Transformer.

on long sequences [26] and token importance is concentrated in a local receptive field of around 50 tokens around the current time step [35].

An emerging body of research suggests that State Space Models (SSMs) can serve as an alternative to Transformers because they are able to capture dependencies in extremely long sequences, while being more computationally efficient and parallelizable [14]. While still falling into the category of autoregressive sequence models, the underlying linear time-invariant dynamical system of SSMs allows the efficient processing of sequences using parallelizable convolution operators with the Fast Fourier Transform (FFT) [7], with $\mathcal{O}(L \log L)$ complexity, where $L$ is the length of the sequence. Moreover, retention of past information over long sequences, up to thousands of steps, can be ensured by deriving recurrent update rules by borrowing ideas from online function approximation [3, 12]. SSMs have recently outperformed Transformers on long-range dependency benchmarks by a large margin [37]. Despite their success on long-range classification tasks, SSMs have not yet completely matched Transformers as an off-the-shelf sequence model for general language modeling tasks [10].

Recent findings suggest that Transformers and SSMs are complementary models for the purpose of language modeling [28]. In this work, we propose an architecture that integrates a strong local attention-based inductive bias with the long-term context modeling abilities of SSMs into a single layer, that we call *Block-State Transformer* (BST). Our model is able to process long input sequences, while still incorporating an attention mechanism to predict next tokens. BST is fully parallelizable, scales to much longer sequences, and offers a $10\times$ speedup compared to comparable Transformer-based layers.

In every BST layer, an SSM takes the entire sequence as input and maps it into a "context" sequence of the same length. The SSM sublayer takes advantage of FFT-based convolutions. This sequence of context is then divided into blocks of equal size, i.e. window length ($W$), and each context block is then fed to a Block Transformer layer, that attends to the subsequences of size $W$ as defined in [21]. The block of input token embeddings are then cross-attended to the corresponding block of context states; see Figure 1. Note that by introducing SSMs as a means of contextualization, we completely remove the need for sequential recurrences and we are able to run our hybrid SSM-Transformer layer fully in parallel. The resulting runtime complexity can be expressed as the sum of $\mathcal{O}(W^2) + \mathcal{O}(L \log L)$, where the first term represents the time complexity of the Transformer sublayer, while the second term represents the time complexity of the SSM sublayer. This is a major improvement over $\mathcal{O}(LW)$ of Block-Recurrent Transformer, so long as hardware to support parallel computation is available. Moreover, due to hardware imposed restrictions, the runtime complexity of the SSM on a full sequence is comparable to that of Block Transformer on a block of tokens, which further implies the absence of a speed bottleneck in the BST layer, empirically validated for sequences containing hundreds of thousand of tokens. This is evident by observing that the bottom-most two lines on the left of Figure 4 are almost overlapping.

## 2   Related Work

This work is primarily related to two branches of recent research: (i) combining local attention with recurrent networks in order to extend their capacity to capture long-range dependencies, beyond the length of the attention window size, and (ii) State Space Models (SSMs) which describe sequences via linear dynamical systems whose outputs can be computed in parallel. Block-Recurrent Transformer (BRECT) [21] uses a recurrent memory mechanism to extend the theoretical context length of the Transformer. In the recurrent unit of the BRECT cell, the updates made to the "recurrent state vectors," are extracted by employing a cross-attention mechanism over a block/window of input token embeddings. Different from their work, we use linear state space models instead of recurrent cells to maintain context states. We also conduct a more extensive exploration of maintaining and updating context states. Earlier works that augment transformers with a non-differentiable external memory include the Memorizing Transformer [42]. Transformer-XL [8] was an early work that combined recurrent memory with Transformers. Our work can be seen as a continued evolution of those models incorporating state-of-the-art recurrent memory models inspired by SSMs.

State space models can be considered as linear RNNs [12]. This simplicity facilitates their analysis and even enables analytical derivation of recurrent weights for optimally representing arbitrarily long sequences. The linear property also allows the recurrence to be unrolled and parallelized during training and inference [14]. Our work combines these state-of-the art models, enabling Transformers to leverage theoretically infinite context.

Other works have attempted to replace Transformers, and their attention mechanism with SSMs [28, 27, 10, 30], however despite recent progress, the performance achieved by the Transformer architecture remains unparalleled in language. Nevertheless, SSMs are able to capture longer range dependencies than Transformers in both theory and practice, while also being highly parallelizable [7, 11]. We therefore elect to combine the best aspects of SSMs and Transformers into a single model. The idea of communication across blocks, similar to GSS [28], was later implemented by MEGA [27], through an Exponentially Moving Average (EMA) update rule instead of SSMs[2]. However, both GSS and MEGA use a single-head Gated Attention Unit (GAU) [20]. MEGA further mixes layer inputs, GAU outputs and EMA outputs via two gating mechanisms. Our method uses a simpler architecture to mix signals from local attention and SSM outputs via cross-attention, allowing us to plug any out-of-the-box SSMs or attention layers. Further, we investigate three ways to mix SSM signals with attention as outlined in Section 3.3.

## 3   Method

We consider the problem of next token prediction via a decoder-only language model. This seemingly simple pretext task has led to spectacular progress in language understanding [9, 2, 29]. During training, the decoder takes in a sequence of length $L$ of tokens embeddings and is tasked to generate the next token at every step in the sequence.

We start by a brief review of SSMs that are essential for understanding the Block-State Transformer layer (3.1). Our full Block-State Transformer architecture is outlined in Section 3.2. Section 3.3 describes three approaches for integrating SSM states into the attention mechanism. Important implementation details are described in Section 3.4.

### 3.1   State Space Preliminaries

State space models can be divided into two categories:

**State Spaces: Structured Kernels**   S4 [14], S5 [34], S4D [15], DSS [16], follow a structured initialization of the convolutional kernel by unrolling a linear time-invariant (LTI) dynamical system of the following form:

$$
\begin{aligned}
x_k &= \mathbf{A}x_{k-1} + \mathbf{B}u_k \,, \\
y_k &= \mathbf{C}x_k + \mathbf{D}u_k \,.
\end{aligned}
\tag{1}
$$

---

[2]The authors in [27] show a mathematical form of EMA that has a state transition and also derive a convolution kernel to efficiently compute EMA similarly to S4.

The system is parameterized by a state matrix $\mathbf{A} \in \mathbb{R}^{N \times N}$, vectors $\mathbf{B} \in \mathbb{R}^{N \times 1}$, $\mathbf{C} \in \mathbb{R}^{1 \times N}$, and $\mathbf{D} \in \mathbb{R}^{1 \times 1}$, the SSM maps a 1-D input signal $u_k$, to a 1-D output signal $y_k$. Internally, the SSM projects the input signal to an $N$-D representation state $x_k$, before mapping it down to a scalar using the $\mathbf{C}$ matrix. The term $\mathbf{D}u_k$ can be thought of as a skip connection and will be omitted for the remainder of the discussion for convenience. The output of the above recurrent equation, $y_k$, can be computed as a discrete convolution, by realizing that the recurrence can be explicitly unrolled:

$$\text{Let} \quad x_{-1} := \vec{0} \, ,$$
$$y_k = \sum_{j=0}^{k} \mathbf{C}\mathbf{A}^j \mathbf{B} \cdot u_{k-j} \, . \tag{2}$$

The $\mathbf{C}\mathbf{A}^k\mathbf{B}$ entries are collected to create the SSM kernel $\mathbf{K} \in \mathbb{R}^L$, and the convolution could be expressed as:

$$\mathbf{K} = (\mathbf{C}\mathbf{B}, \mathbf{C}\mathbf{A}\mathbf{B}, \ldots, \mathbf{C}\mathbf{A}^{L-1}\mathbf{B}) \, ,$$
$$y_k = \sum_{j=0}^{k} \mathbf{K}_j \cdot u_{k-j} \, , \quad y = \mathbf{K} * u \, . \tag{3}$$

Given an input sequence $u \in \mathbb{R}^L$, it is possible to compute the output $y \in \mathbb{R}^L$ sequentially through the recurrence in Equation (1). While this property is useful for autoregressive decoding, sequential computation is prohibitively slow to train with long inputs and, instead, the convolution from the Equation (3) can be used to compute all elements of $y$ in parallel. This is done via Fast Fourier Transform (FFT) [7], provided we have already computed $\mathbf{K}$.

Additional inductive biases have been imposed on SSMs by *analytically deriving* closed-form expressions for the matrices $\mathbf{A}$ and $\mathbf{B}$ using the HiPPO framework [12]. In this framework, the state $x_t$ represents the coefficients of polynomials that approximate the sequence $u_t$.

**Explicitly Parameterized Filters**   In contrast to structured kernels, one can parameterize the convolution kernel, as trainable weights and optimize them, $\bar{\mathbf{K}} \in \mathbb{R}^L$. However, this would result in poor performance unless certain types of regularization are applied to the kernel. [11] simply makes use of squashing the kernel weights, and subsequently applying a smoothing technique. Trainable kernels are also used in attention-free alternative models to Transformers, such as Hyena [30], which involves exponentially decaying the weights along the kernel:

$$\bar{\mathbf{K}}_t = e^{-\alpha t} \cdot \big(\mathsf{FFN} \circ \mathsf{PositionalEncoding}\big)(t) \, , \tag{4}$$

where $\bar{\mathbf{K}}_t$ is an entry in the filter at location $t$, and FFN is a feed-forward network used for decoupling the parameter count from the seuqnece length.

### 3.2   Block-State Transformer (BST) Layer

We now introduce the Block-State Transformer layer, which combines SSMs with Block Transformers. At each training iteration, a sequence of $L$ tokens, is sampled from a longer document. The tokens are then embedded and fed to the model. Our model consists of a stack of Block-State Transformer layers. Each BST layer optionally includes an SSM sublayer that is responsible for providing long-range context to the Block Transformer layer, which operate similarly to a Block-Recurrent Transformer (BRECT) cell. The SSM sublayer takes the sequence of token embeddings from the previous layer as input, and produces a sequence of the same length $L$ as the output.

The output of the SSM is contextually encoded, meaning that entries at every time-step, potentially include information about all the time steps preceding elements in the sequence. We collect a number of "context states," $S$, from the context sequence, and we set $S \ll L$. In order to prevent the model from accessing future information, we only allow the model to access context states that precede the current token. Various ways to gather context states from the context sequence are discussed in section 3.3 in detail.

The context states are fed to the Block Transformer, in place of what was referred to as "recurrent state vectors" in Block-Recurrent Transformer [21]. The subsequent operations, shown on the right side of

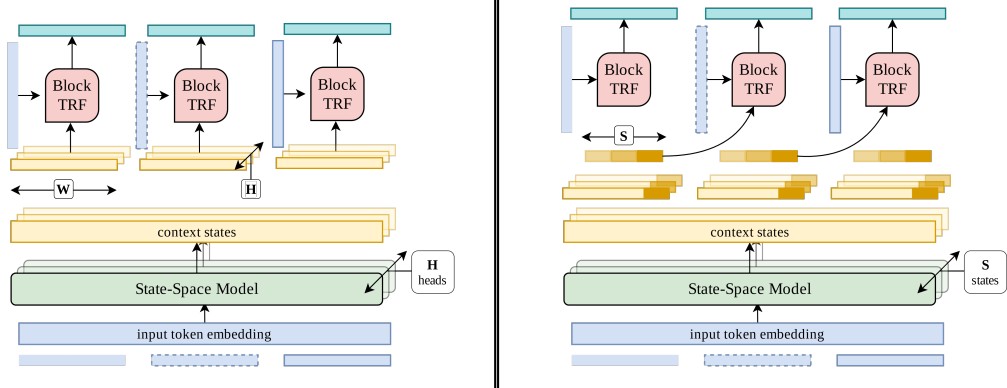

Figure 2: Summarizing our approaches. The left side shows the cases where the SSM is required to output Multi-Head (**MH**) contexts. On the right Multi-Filter (**MF**) approach is depicted where the last entries from the previous window are concatenated into a set of context states of size $S$. Dashed lines represent the current block.

Figure 1, are kept unaltered, except that we no longer need to run the recurrent unit of the BRECT cell since we are maintaining the context via an SSM. In addition to the context states, the Block Transformer also receives a block/window of length $W$ of token embeddings as input, which are cross-attended to the context states. The output of the cross-attention operation is then concatenated with that of self-attention over the input embeddings, followed by a simple projection.

In addition to the ability of SSMs to retain information over longer time horizons compared to Transformers and RNNs, using the SSM to maintain context states as a replacement for recurrent cells makes for a more computationally efficient layer. Removing recurrence by integrating SSMs into Transformer layers, allows the Block-State Transformer layer to be fully parallelizable, whereas the Block-Recurrent architecture processes blocks of tokens sequentially using a for loop.

### 3.3 Context States

Although the latest SSM output technically contains information about the entire sequence, retrieving individual tokens from only the final state may not be feasible. To compensate, we concatenate a sequence of states, corresponding to the latest block of tokens. This is also analogous to the approach taken by BRECT. This representation ensures *retrievability* and ease of access, through *redundancy*. It is redundant because adjacent states are highly correlated, however this also makes it possible to easily recover the current block of tokens, if necessary.

In our approach, the context states are constructed from the output of the SSM and fed to the attention heads of the Transformer. These context states can be constructed in various ways. To guide these design decisions we consider each of the below proposed schemes as introducing *retrievability* at the cost of *redundancy*. The shape of the output of a single SSM layer is $(B \times L \times D)$, where $B$ is the batch size, $L$ is the number of the tokens processed, and $D$ is the embedding dimension. When doing cross-attention in the Transformer cell with $H$ different heads, this tensor needs to be transformed into a context tensor of shape $(B \times S \times D \times H)$, where $S$ is the number of context states; we usually set $S \ll L$ and $S = W$ similar to Block-Recurrent Transformers (BRECT).

We now discuss the three different approaches that we evaluate to generate a context tensor for each block sequence:

**SH: Single-Head**  The first approach constructs the context tensor by sequentially concatenating the $S$ states from the SSM with a single filter (each of size $D$). Note that because the SSM captures information from preceding blocks, the context state also captures information about blocks that preceded the current block. The resulting context vector is highly *retrievable* and *redundant*, as defined above. As in typical Transformers, fully connected layers are used to project each context vector to $H$ different heads of size $D$. Note that in the cross-attention operation, context states that correspond to future tokens from the current block need to be causally masked out. In this case we set $S = W$, and we pick the window of SSM outputs that correspond to the current block, and a

triangular mask is used to implement causal masking of context states. This approach is shown in Figure 1.

**MH: Multi-Head**    This approach differs from Single-Head (SH) in that here the SSM is tasked to generate a separate output for different heads. We use separate $[\mathbf{C}_1, \mathbf{C}_2, ..., \mathbf{C}_H]$ matrices, to produce context states that are fed to the attention heads. This enables the SSM to extract complementary features from the summarized history. The conceptual difference is that the $\mathbf{C}$ matrix, from Equation (1), has direct access to the full memory state of the SSM ($x_k$), that in theory could be thought of as a compact representation of the history, before it gets mapped down to a scalar. The Multi-Head (MH) approach is illustrated on the left side of Figure 2. Because the $H$ different $\mathbf{C}$ matrices may extract complementary information, the context vector constructed by this method is theoretically less *redundant* compared to the single-head method described above.

**MF: Multi-Filter**    In this approach the SSM sublayer produces $S$ context states, which we set to be independent from $W$. This is done by convolving the sequence of embeddings with $S$ different kernels/filters. The output of each convolution operation, corresponding to a specific filter, is a tensor of shape $(B \times L \times D)$. After convolving the input with all the filters, the context states of size $D$ that correspond to the *last token* from the previous window are stacked together to make a $(B \times S \times D)$ tensor. Feed forward networks are then used to lift this tensor to different heads, $(B \times S \times D \times H)$. Different from the previous two approaches, the context is formed by taking only the *last $S$* context states, from the previous window, outputted by the $S$ SSMs. The context is less redundant because it no longer consists of adjacent SSM states. Since the context is taken from the entries of the previous window, cross-attention masking is no longer required, as shown on the right of Figure 2.

The memory states of the Multi-Filter (MF) approach is least *redundant*, while Multi-Head (MH) strikes a middle ground, and Single-Head (SH) has the most redundancy. The incorporation of *redundancy* in these approaches aims to facilitate *retrievability* of the most recent context captured by the SSM, albeit at the expense of potentially inefficient *utilization* of the network capacity. The last approach attains highest *utilization*, as the cross-attention is done in the space of unique features extracted by specialized filters.

### 3.4   Implementation Details

**Context IDs & Positional Embedding**    To allow distinction between the entries supplied to the attention mechanism, a positional embedding is commonly added to the inputs. When using the Multi-Filter (MF) approach, the collected context states correspond to different features extracted from the sequence, hence we add a set of unique learned "context IDs" to the context states, before using them as input to cross-attention. However, in the cases where the context states correspond to different time-steps along the sequence, namely Single-Head (SH) and Multi-Head (MH) approaches, inherent positional encoding is incorporated into the context states, due to the incremental nature of convolutions; as such, we find the addition of context IDs to be unnecessary. We also realize that we do not need to add global positional bias to the token embeddings, and use a T5-style relative position bias [32] instead, as the SSM does also encode positional information into the context.

**Down-sampling**    Consistent with findings in [28], we find FFT operations to be the main source of bottleneck when training SSMs on TPUs. We project the input embeddings to a lower-dimensional space, that is a quarter of embedding size in our experiments, this reduces the required total number of FFTs by a factor of $4$. The output of the SSM, i.e. the context states, are later lifted to the original embedding size before being passed to the Block Transformer.

## 4   Results

Our results are presented in Table 1. We conduct experiments with BST on three different datasets, PG19, arXiv and GitHub, allowing us to test our method on a suite of varying documents lengths composed of English texts, latex scientific articles and source code.

**PG19** dataset is from a large collection of full-length books from Project Gutenberg [31]. All extracted 28,602 books were published prior to 1919 and contain 6,966,499 English language words. When tokenized, each PG19 book has between 50k-100k tokens. PG19 has become a popular

Table 1: Perplexity of each model. The results for XL:2048, Slide:12L and BRect:fixed:skip are from [21] by converting $\log_2$ of perplexity to raw perplexity. GSS-Hybrid-L performance was taken from [28]. Results with $\pm$ are average scores and error bars of runs with three different random seeds. For a *smaller computational budget*, BST provides a small perplexity improvement compared to BRect on PG19 and GitHub. For the same computational budget, BST outperforms GSS-Hybrid-L across datasets by 1.5% to 4%.

| Model | eval seq. length | window length | number params | TPUv4 hours (k) PG19/arXiv/GitHub | PG19 | arXiv | GitHub |
|---|---|---|---|---|---|---|---|
| Slide:12L | 4096 | 512 | 190M | 0.5 / 0.5 / 1.8 | 12.12 | 2.69 | 2.28 |
| Trsf-XL:2048 | 2048 | 2048 | 190M | 0.8 / 0.8 / 3.0 | 11.96 | 2.48 | 2.01 |
| BRect:fixed:skip | 4096 | 512 | 196M | 0.8 / 0.8 / 3.0 | 11.55 $\pm$1.1 | **2.36** | 2.04 |
| BST:SH:S4 | | | 202M | 0.5 / 0.5 / 1.8 | 11.57 $\pm$1.1 | 2.51 | 2.14 |
| BST:MH:S4 | | | 218M | 0.8 / 0.8 / 1.8 | 11.60 $\pm$1.1 | 2.52 | 2.15 |
| BST:MF:S4 | | | 217M | 0.5 / 0.5 / 1.8 | 11.63 $\pm$1.2 | 2.48 | 2.07 |
| BST:SH:unstruct | | | 206M | **0.5 / 0.5 / 1.8** | **11.52** $\pm$1.1 | 2.49 | 2.09 |
| BST:MF:unstruct | | | 221M | **0.5 / 0.5 / 1.8** | 11.56 $\pm$1.2 | 2.44 | **2.03** |
| GSS-Hybrid-L | 4096 | 512 | 373M | 0.8 / 0.8 / 1.8 | 10.52 | 2.51 | 1.88 |
| BST:SH:S4-L | | | 366M | 0.8 / 0.8 / 1.8 | 10.47 | 2.49 | 1.86 |
| BST:MF:S4-L | | | 383M | 0.8 / 0.8 / 1.8 | 10.52 | 2.46 | 1.84 |
| BST:SH:unstruct-L | | | 371M | 0.8 / 0.8 / 1.8 | **10.37** | 2.46 | 1.85 |
| BST:MF:unstruct-L | | | 388M | 0.8 / 0.8 / 1.8 | 10.42 | 2.41 | **1.83** |

benchmark for measuring progress on long-range language modeling performance. We report the "test" split evaluation performance.

**arXiv** dataset is a corpus containing scientific and technical articles on the subject of Mathematics [42]. The arXiv dataset contains latex source code as well as items such as theorems, citations, definitions that are referenced and discussed over long ranges of text. Using the same vocabulary as in [42] and [21] for a fair comparison, many special characters are broken up into small subwords. As a result, the number of tokens per paper in the arXiv dataset is approximately equal to the number of tokens per book in PG19. We report perplexity on "test" split.

**GitHub** dataset [42] is the largest of the three datasets and was assembled by extracting GitHub code repositories with open-source licences. Files were filtered to only contain the following programming languages: C, C++, Java, Python, Go and Typescript. While code files are relatively small, there are many import dependencies between each file. By traversing the directory tree and concatenating all code files along the path, a single document that preserves a repository's structure and dependencies is created. We report performance on the "validation" split.

For a fair comparison with the baselines, we keep the vocabularies consistent as used by [21] and [28]. Specifically, we used a pretrained T5 vocab with 32k tokens for PG19 [33] and LaMDA vocab with 32k tokens [39] for both arXiv and GitHub datasets. Due to the long training times and large number of experiments, we only provide error bars for the PG19 ~200M parameter models by running our models with three different random seeds. BRect:fixed:skip error bars are from [21].

### 4.1 Comparing our Baselines and Models

We experiment three different types Block-State Transformer (BST) models: BST-SH, BST-MH and BST-MF as described in Section 3.3. Our models do not use global learned positional embeddings but encode positional awareness with an SSM at the first layer, right after the word embedding layer. We organize models into two groups: (i) *fixed window size* have either a 512 or a 2048 token training window size; and (ii) *fixed parameter count* have either a ~200M or ~400M total parameters. We run experiments with two types of SSMs:

BST:{SH,MH,MF}:S4 encode long context using a Structured State Space Model (S4) [16]. As described in Equation (3), S4 kernel matrix $\mathbf{K}$ is compiled from matrices $\mathbf{A}$, $\mathbf{B}$ and $\mathbf{C}$ and is independent of the length of the input evaluation sequence length. We show that the structured parameterization of $\mathbf{K}$ allows our BST models to generalize to longer lengths. We refer the reader to

section 4.2 for results on length generalization. We only run one BST:MH using S4 since the model requires 8% more parameters while performing on par with the faster BST:SH variant. BST:MF also has 8% more parameters but performs better on arXiv and GitHub compared to SH. Interestingly, SH performs better than MF on the PG19, a dataset where local context is more important to predict the next token compared to arXiv and GitHub. We posit that this is likely due to the ability of the SH model to *retrieve* the most recent context captured by the SSM.

BST:{SH,MF}:UNSTRUCT are based of unstructured parameterized convolution filters, inspired by the Hyena Hierarchies [30] convolutional kernel. We exclude the utilization of the multiplicative gating mechanism employed in Hyena Hierarchies and solely apply the regularizations implemented on the parameterized kernel, denoted as $\bar{\mathbf{K}}$ in Equation (4). This formulation has two important advantages over S4: (1) the $\bar{\mathbf{K}}$ kernel does not need to be recompiled, allowing speedups when using multiple filters; (2) $\bar{\mathbf{K}}$ has more free parameters because it is no longer restricted by $\mathbf{A}$, $\mathbf{B}$ matrices in equation 3, potentially providing richer representations that can explain the improved perplexity scores over S4 variants. Nonetheless, UNSTRUCT kernel $\bar{\mathbf{K}}$ relies on learned positional encoding which makes the method less extendable to larger length sequences at inference..

We compare the Block-State Transformer to four different baselines:

TRSF-XL:2048 [8] is a Transformer with a training window size of 2048. As expected, increasing the window size improves perplexity, especially on the arXiv and GitHub datasets. However, this model performs worse than BST:SH:HYENA on PG19 and is much slower, bottlenecked by the attention layer on higher sequence lengths.

SLIDE:12L [21] This model is almost identical to TRSF-XL:2048. It uses however a sliding window of size 512 over a segment of 4096 tokens. The sliding window is differentiable over two blocks, while TRSF-XL does not backpropagate through the cached keys and values from the previous window. This simple baseline is closest in terms of training speed to BST:SH. The perplexity scores show that integrating a representation of the past, as with BRECT and BST, positively impacts LM performance.

BRECT:FIXED:SKIP [21] is the strongest performing and fastest Block-Recurrent Transformer architecture in [21]. This architecture is very similar to SLIDE:12L. There is however a sequential recurrent "skip" configuration, a simple linear layer gating mechanism that combines current block hidden representation with past information from the previous blocks.

GSS-HYBRID-L [28] is the closest SSM-Transformer hybrid model that was tested on long-range language modeling tasks. GSS-HYBRID-L is based on the Diagonal State Space (DSS) [16]. DSS and S4 are similar in performance and architecture, only differing on the initialization of the kernel $\mathbf{K}$ [15]. [16] further improves on DSS for LM tasks by introducing a Gated State Space version called GSS, which performs better on PG19, arXiv and GitHub. Unlike our method, GSS-HYBRID-L does not directly integrate SSMs states into the attention mechanism but only interleaves 32 GSS layers with Transformer layers. It must be noted that the GSS-HYBRID-L scores were obtained after grid searching over four learning rates $\{6.4, 3.2, 1.6, 0.8\} \times 10^{-3}$ and used a different learning rate and weight decay for the SSM layer and the Transformer layer to avoid training instabilities. In our experiment, we did not use grid search and used the same learning rate for all layers. BST results demonstrate that integrating SSM states into the Transformer attention provides larger benefits than interleaving SSM and attention layers as in GSS-HYBRID-L.

**Fixed compute budget.**    As seen in Table 1, we track the exact amount of compute in TPUv4 hours that was spent training each model. The training TPUv4 hours for SLIDE:12L, TRSF-XL:2048, BRECT:FIXED:SKIP and GSS-HYBRID-L were taken from [28]. The TPUv4 hours metric measures the compute cost of training models. For our experiments, we align our training times with GSS-HYBRID-L for a fair comparison. Smaller parameter models all have 12 layers, 8 heads of size 128, embedding vectors of size 1024, an MLP with a hidden layer size of 4096 with ReLU activation functions. For larger BST models, we double the intermediate layer size from 4096 to 8192 and increase the number of attention heads to 12.

**Training details**    We use the same training setup as [21] and we perform our experiments using the Meliad library[3] in JAX/Flax [1, 17]. We use the Adam optimizer [25] and a batch size of 32

---

[3]https://github.com/google-research/meliad

and a sequence length $L$ of 4k for training. Using a structured SSM's recurrence (such as S4) in the first layer allows us to extend the positional encoding to various lengths at inference. Smaller BST models have Block-State layer integrated in Transformer layers {1, 7, 9} and larger BST models at layers {1, 5, 7, 9}. Since our datasets contain long documents, it is possible to train on larger sequence lengths $L$. Training on 4k sequence lengths allows us to test length generalization since the convolution kernel $\mathbf{K}$ in Equation (3) can be extended to any sequence length $L$. However, since we show in Section 4.2 that our model works well when extended to unseen lengths, we did not find it necessary to run expensive experiments with higher sequence lengths. For the MF model variants, we lower the SSM state dimension $D$ by an additional factor of two to improve FFT efficiency. The state dimension reduction has negligible impact to perplexity. The MF models have $S = 32$ filters while the larger MF models have $S = 64$ filters.

## 4.2 Evaluating Length Generalization capabilities

We present our length generalization analysis and report perplexity in Figure 3. Our models and baselines all have ~400M parameters, are trained on a sequence length of 4k and tested on sequences with *lower* and *higher* sequence lengths of {512, 16k, 65k}.

We notice that all models have similar perplexity for sequence lengths of 512. Both BST:SH:S4-L and GSS-HYBRID-L generalize well on 16k and 65k sequence lengths for PG19 and GitHub. For arXiv, GSS-HYBRID-L and BST:MF:UNSTRUCT-L perplexities increase drastically, potentially due to noise in the arXiv dataset (as indicated by variation in perplexity metric over time). [28] also reported that larger GSS models had difficulty generalizing to higher lengths. Interestingly, for arXiv again, BRECT:FIXED:SKIP-L performs very well at higher sequence lengths. We hypothesize that the Block-Recurrent model's access to the entire past during training, via a non-differentiable cache of representations across sequences, helps retain a "memory" of dependencies between key items in an arXiv article allowing the model to access past symbols, definitions, theorems or equations beyond the 4k training sequence length. We also note that BST:MF:UNSTRUCT-L and BRECT:FIXED:SKIP-L outperform other methods on PG19 up to a sequence length of 16K. Perplexity performance on PG19 is perhaps less reliant on long term relationships between tokens, which can explain the performance of models that have no explicit built-in mechanisms for length generalization.

The analysis also allows us to draw a clear distinction between *structured* and *unstructured* SSMs integrated in hybrid architectures. As previously mentioned in Section 3.1, SSMs such as DSS and S4 use a structured kernel $\mathbf{K}$, built from learned matrices $\mathbf{A}$, $\mathbf{B}$ and $\mathbf{C}$ for any sequence length $L$ in Equation 3. Since $\mathbf{K}$ is extendable to any arbitrary sequence length $L$, both BST:SH:S4-L and GSS-HYBRID-L have a build-in mechanism for length generalization that the unstructured BST:MF:UNSTRUCT-L model does not. BST:MF:UNSTRUCT-L performs best on the training sequence of 4K and is on-par for 512 with perplexity increasing for unseen 16K and 65K sequence lengths. **BST:SH:S4-L has by far the best perplexity for 65K sequence lengths on PG19, GitHub and arXiv.** Similarly to [21], we also notice that perplexity improves when we extend context window (sequence length) for PG19 and GitHub.

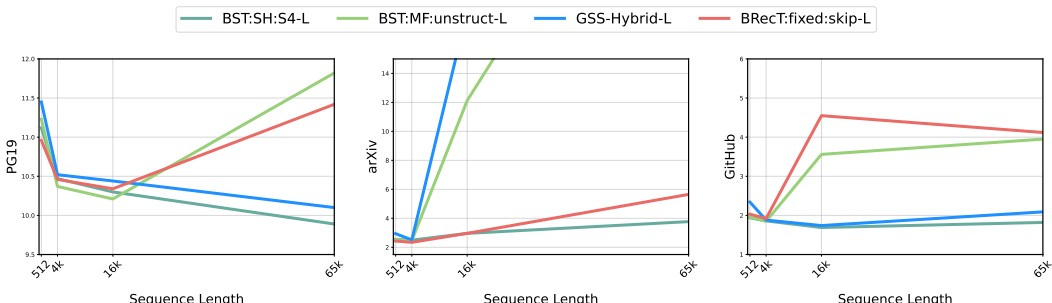

Figure 3: Length Generalization for sequence lengths {512, 16k, 65k} on PG19 (left), arXiv (middle) and GitHub (right). BST:SH:S4-L generalizes better than other baselines, including GSS-HYBRID-L that uses GSS, a structured SSM. GSS-HYBRID-L numbers are from [28].

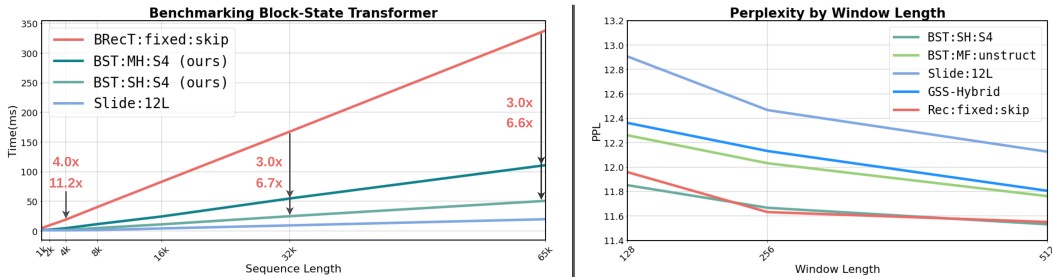

Figure 4: **Left**: The forward-pass computation time of a BST layer is compared against a layer of BRECT and SLIDE:12L. These experiments were executed on GPU, to demonstrate and exploit the parallelizability of BST layers. BST:SH is 6-11× faster than BRECT while BST:MH is 3-4× faster. **Right**: Perplexity of the trained models using different window lengths. The figure shows that increasing the training window length results, as expected, in better perplexity scores. We find however that both BST:MF:HYENA and BRECT:FIXED:SKIP are the least impacted by decreasing window lengths.

### 4.3 Efficiency

The improvement over Block-Recurrent Transformers, with time complexity of $\mathcal{O}((W^2 + S^2 + 2SW) \cdot L/W) \approx \mathcal{O}(L \cdot W)$, follows from the ability to run the Block Transformer's cells in parallel. The time complexity of the Block-State Transformer layer is comprised of the time complexity of the state space model sublayer, $\mathcal{O}(D \cdot L \log L)$, in addition to the time complexity required to execute the Transformer over the given context chunks (blocks) in parallel, $\mathcal{O}(W^2)$.

In spite of the superlinear growth of the SSM sublayer, our experiments indicate that significant performance improvements, up to a factor of 6, remain evident for sequences as long as 65k tokens, the point at which hardware saturation began to occur. When using a structured SSM, the computational complexity is closely tied to the internal memory state size of the SSM, $N$ – specifics may vary depending on the exact type of the SSM. We set $N = 16$ when reporting performance. Left side of Figure 4 shows the results of benchmarking the forward-pass of a Block-State Transformer layer on GPU. Our proposed layer runs almost 6-11× faster than Block-Recurrent Transformers (including recurrent units), and yields comparable performance to a SLIDE:12L layer, i.e. BRECT without the recurrence. At 4k sequence length, which is mostly used during training, BRECT layer runs almost 15× slower than SLIDE:12L with the same window size. We manage to reduce this gap to less than 2× with BST layer. To reflect a realistic model, for these experiments we use a fixed window length of 128, an internal state size of 16 for the SSM, and 16 heads. Moreover, to highlight the performance gains that are only due to parallelization made possible by our framework, we use same embedding size as input to the SSM, which is 512. Note that we use the vanilla implementation of FFT and inverse FFT operations provided by JAX [1]. However, we believe that the speed of our method can be further improved with recent and faster hardware-specific I/O-aware implementations introduced in other auto-diff frameworks.

## 5 Conclusion

We have introduced a model that combines the attention mechanism of Transformers with the long-range memory mechanism, and parallelism afforded by State Space Models. We explored several memory state variants that make different trade-offs between *redundancy* and *retrievability*. Experiments show that our model can minimize perplexity on par with and often improves upon recent competing baselines, while achieving up to more than 10× speedups at the layer level, provided there is hardware support to fully take advantage of parallelism. This is an appealing property for scaling up BST which makes the addition of SSMs into Transformers computationally appealing. We show that integrating SSM states into the Transformer attention provides larger benefits than simply interleaving SSM and attention layers. Finally, we show that the model generalizes to longer sequences than it was trained.

## Acknowledgments

We would like to thank Caglar Gulcehre and Albert Gu for helpful discussions and support with the S4 codebase. We would also like to express our gratitude to Delesley Hutchins for providing valuable guidance throughout the project, as well as Xavier Garcia and Courtney Paquette for their careful review of the manuscript, where they identified and rectified several errors.

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

## A    Limitations

While BST's SSM layer allows the model to unroll and parallelize the recurrence that models long-term context between blocks of tokens, the SSM variants are reliant on efficient FFT operations. We have found that the FFT operation is an important speed bottleneck on TPUs that needs to be resolved to better scale BST to many layers and larger models. While we are still investigating the reasons, we found that JAX FFT was $4\times$ faster on GPUs. Further, new SSM variants such as S5 [34] bypass FFT operations using a binary associative operator[4]. Our implementation is modular enough that we can simply plug in S5 or use other FFT implementations.

One of our assumptions is that BST's SSM layer is able to capture the right long-term dependencies for each block. The SSM recurrence at step $T = t$ provides a summarized representation of previous steps for $T = 0$ to $T = t$. However, a single vector representation may not be powerful enough to support all important long-term dependencies. Despite the perplexity improvements on long-range language modeling tasks, this assumption needs to be tested on other long-range classification tasks such as Long Range Arena [37] as well. It is possible that our model can perform better if we feed to the attention layer $k = W$ SSM representations that are chosen by a top-$k$ retrieval operation, similar to the one in Memorizing Transformer [42].

## B    More detailed comparisons with existing baselines

This section provides the reader with a more in-depth comparison with similar architectures. We cover BRECT [21] in Section B.1 and GSS-HYBRID [28] in Section B.2.

### B.1    Comparison with Block Recurrent Transformer (BRECT)

The Block Transformer sublayer (i.e SLIDE:12L) processes keys and values from the previous window stored in a differentiable cache. This is implemented similarly to the sliding window attention pattern suggested in [21] and was originally introduced by Transformer-XL [8]. Using a causal mask, at every token inference step, the attention mechanism is applied to blocks of tokens of size $W$ and is partially extended to the cached keys and values from the previous block with the sliding window. BRECT, as explained in [21], uses a non-differentiable cache that is carried from one sequence of size $L$ to the next[5]. The last recurrent states of a sequence are stored in a non-differentiable cache and fed to the next training step on the following sequence in the document as a warm-start. We do not pass such a representation, since to compute the output of the convolution, we need access to the whole sequence. We believe that this is one advantage that BRECT has over our method, especially for very long examples that split into ordered sequences of length $L$, since the cache carried from one sequence to the next can provide very useful long-range information and (weak) access to the whole past. Since we need the whole sequence to compute SSM states, history beyond $L$ may be lost in the process. We believe that BST can further be improved by adding non-differentiable sequence cache for very long documents.

While in other architectures, the history between blocks of tokens is not modeled, both BST and BRECT use a mechanism to model previous block context. The authors of BRECT experiment with various sequential gating mechanisms to condense the information from past blocks. With BST, we use SSM to provide context from previous blocks to the current block as explained in Section 3.2.

### B.2    Comparison with the Transformer GSS-HYBRID

GSS-HYBRID [28] is a SSM-Transformer hybrid architecture that we first describe in Section 4.1. The architecture is significantly different from BST. GSS-HYBRID is primarily composed of Gated State Space (GSS) layers and has a few interleaved Transformer layers at every 4th layer starting with the 2nd layer. BST on the other hand is mainly composed of Block Transformer layers and has Block-State Transformer layers at positions $\{1, 7, 9\}$ for the $\sim$200M model and $\{1, 5, 7, 9\}$ for the $\sim$400M model. Our hybrid does not stack SSM and Transformer layers like the GSS-HYBRID but rather replaces the recurrence in BRECT with an SSM such as S4. In BST, the SSM generates states

---

[4]In JAX, this is equivalent to using $jax.lax.associative\_scan$.

[5]In our work and in [21], a document is split into multiple sequences of size $L$ and each sequence is split into multiple blocks of size $W$

for each Block Transformer representations and we then use cross-attention to mix the states and the self-attention outputs. The authors in [28] initially built GSS, a gated version of DSS [16], to (1) reduce SSM parameter dimensions, (2) stabilize training of the SSM and (3) allow better length generalization. However, when experimenting with SSMs such as S4 or DSS, we found that the gating was not necessary to achieve all three objectives stated above. We decided that using GSS's Gated Attention Unit [20] was therefore not needed when integrating SSM states into the attention mechanism. We also reiterate that the authors in [28] used hyperparameter search to get the best performance while we did not.

## C  Scaling Experiments

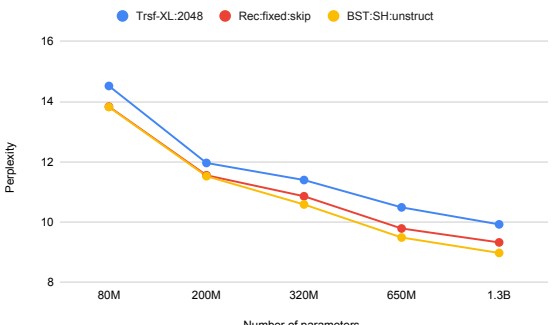

Figure 5:
Scaling properties on PG-19.

Yellow: (BST:SH:UNSTRUCT)
12-layer Block-State Transformer.

Red: (REC:FIXED:SKIP)
12-layer Block-Recurrent Transformer.

Blue: (TRSF-XL-2048)
13-layer Transformer-XL.

In this section, we compare how BST scales compared to Transformer-XL with $4\times$ the window size and BRECT. In Figure 5, we see that at lower scales, from 80M to 200M, BRECT and BST have very similar performances. Beyond 200M, the perplexity performance percentage gap between BRECT and BST increases from 2.5% at 200M paramaters to 4.0% at 1.3B parameters. The perplexity performance percentage gap between BRECT and TRSF-XL is even more pronounced as it starts at 7.6% at 200M parameters to 10.6% at 1.3B parameters.

## D  Long Range Arena Experiments

| MODEL | LISTOPTS | TEXT | RETRIEVAL | IMAGE | PATHFINDER | PATH-X | AVG |
|---|---|---|---|---|---|---|---|
| Transformer | 36.37 | 64.27 | 57.46 | 42.44 | 71.40 | ✗ | 53.66 |
| Linear Trans. | 16.13 | 65.90 | 53.09 | 42.34 | 75.30 | ✗ | 50.46 |
| Reformer | 37.27 | 56.10 | 53.40 | 38.07 | 68.50 | ✗ | 50.56 |
| Performer | 18.01 | 65.40 | 53.82 | 42.77 | 77.05 | ✗ | 51.18 |
| BigBird | 36.05 | 64.02 | 59.29 | 40.83 | 74.87 | ✗ | 54.17 |
| Mega | **63.14** | **90.43** | 91.25 | **90.44** | **96.01** | 97.98 | **88.21** |
| S4D | 60.47 | 86.18 | 89.46 | 88.19 | 93.06 | 91.95 | 84.89 |
| S4 | 59.60 | 86.82 | 90.90 | 88.65 | 94.20 | 96.35 | 86.09 |
| S5 | 62.15 | 89.32 | **91.40** | 88.00 | 95.33 | **98.58** | 87.46 |
| *Methods with chunked input sequences* | | | | | | | |
| BRECT:FIXED:SKIP | 37.29 | 66.14 | 58.76 | 50.41 | 76.33 | 75.89 | 60.80 |
| MEGA-CHUNK | 58.76 | **90.19** | **90.97** | 85.80 | 94.41 | 93.81 | 85.66 |
| BST:SH:S4 (ours) | **61.49** | 87.63 | 90.51 | **91.07** | **95.75** | **95.28** | **86.96** |

Table 2: Performance on Long-Range Arena (LRA). For a fair comparison, we adjust the number of layers and model dimensions on each task so that BST and BRECT have similar number of parameters with S4 and MEGA-CHUNK. BRECT results are from our own runs and all other baselines are from published results.

While the main focus of our research was to demonstrate that hybrid Transformer-SSM models are efficient and perform well on long context autoregressive LM, we also evaluate our method on standard classification task where long range dependencies in a sequence are important to capture. In Table 2, we present our results on the Long Range Arena (LRA) benchmark [38] which incorporates three different modalities including text, images, and mathematical expressions. The LRA dataset also tests models on various sequence lengths from 1K to 16K.

BST:SH:S4 is composed of four BST layers (no BRT layers are interleaved) and two S4 layers on top. We use the same standard block length of 512 for BST and BRT. However, we train BST and BRT on the full sequences (up to 16K for Path-X). We use AdamW as our optimizer [24] with a warmup for the learning rate, where we start from a value of $1e^{-7}$ and increase the learning rate linearly up a specified value $\in \{1e^{-3}, 2e^{-3}, 4e^{-3}\}$ for the first 10% of training. This is followed by cosine annealing for the rest of training down to a value of $1e^{-7}$. All layers are bidirectional, including the S4 layer in BST:SH:S4 as described in [13]. Our weight decay is chosen from {0, 0.05, 0.1, 0.15} and our dropout is chosen from {0, 0.1}. Except for Path-X experiments, we use weight decays $\in \{0.03, 0.05, 0.07\}$ for all parameters except S4D matrices A and B. Also, for Path-X, the initialization range of our discretization time step $\Delta$ for PathX is decreased from $(\Delta_{\min}, \Delta_{\max}) = (0.001, 0.1)$ to $(\Delta_{\min}, \Delta_{\max}) = (0.0001, 0.01)$.

Our results on LRA are very promising and show that, compared to other state-of the art methods that chunk sequences into blocks, BST is able to model long range dependencies. For example, BST outperforms MEGA-CHUNK [27] on four out of six LRA tasks and by 1.5% on the average score. However, BST still needs to improve (perhaps by extending the block size) to catch up to MEGA (without chunks).

# E   Ablation Studies

In the following section, we perform ablations to investigate (1) the placement of a *single* SSM layer in Table 3 in the overall architecture, (2) the effects of the number of SSM layers added in Table 4, and (3) the size $D$ of the SSM state in Table 5. For the ablations, we use the ~200M parameter BST:SH:S4, since it is the fastest model, and assess various configurations on PG19.

Table 3: A single BST at various layer index.

| Layer index | Perplexity |
|---|---|
| 3 | 12.41 |
| 7 | 11.92 |
| 9 | 11.88 |
| 12 | 12.03 |

Table 4: Multiple BST layers at various locations.

| Num layers | Perplexity |
|---|---|
| 2 | 11.69 |
| 3 | 11.57 |
| 4 | 11.21 |
| 5 | 11.20 |

Table 5: Increasing BST's S4 model state size $D$.

| State Size | Perplexity | Step Time |
|---|---|---|
| 8 | 11.95 | ×0.7 |
| 16 | 11.57 | ×1.0 |
| 32 | 11.55 | ×1.8 |
| 64 | 11.54 | ×3.2 |

In Table 3, we experiment adding a single BST layer at layer indices $3, 6, 9, 12$. We notice that a single BST layer with state size $D = 16$ located closer to the middle of the whole Block Transformer stack, at index $= 9$, has the greatest effect on perplexity. This finding is inline with findings in prior work [42, 21].

In Table 4, we test if adding multiple BST layers yields improvements on performance. We start with BST layers with state size $D = 16$ at indices $0, 9$. We follow by adding another BST layer at index 7 for a total of three BST layers and then another at index 5, followed by another at index 12. Adding more BST layers lowers perplexity. However, the results seem to plateau at 5 BST layers. We note also that there is a 3.5% training step time increase for each added layer.

In Table 5, we train our models with different state sizes $D$. For the state size ablation, we use three BST layers at indices $0, 7, 9$. We find that increasing $D$ improves perplexity to the detriment of training speed (step time). For this reason, we chose $D = 16$ for Table 1 BST results.

## F  JAX Implementation of BST

Pseudocode 1 contains a function that implements convolution of multiple filters over the same input sequence using FFT and inverse FFT operations. Pseudocodes 2, 3 and 4 respectively implement context state collection of BST variants: Single-Head (SH), Multi-Head (MH) and Multi-Filter (MF). Finally, Pseudocode 5 runs the Block Transformer sublayer in parallel by feeding the context states to their corresponding block.

```python
"""Unstructured filters and convolutions."""

import jax
from jax import numpy as jnp
from einops import rearrange

win_length = 512    # (w)
seq_length = 4096   # (l)

def get_filters_unstruct(channels):
    """Returns trainable filters and biases.

    Args:
        channels: number of filters.

    Returns:
        h: filter of shape (seq_length, channels, dim)
        b: bias of shape (channels, dim)
    """
    t = jnp.linspace(0.0, 1.0, seq_length)
    h = jnp.exp(- alpha * t) * dense(positional_emb(t))
    b = get_bias()
    return h, b

def multichannel_convolution(u, h, b):
    """Multichannel convolution function.

    Args:
        u: input of shape (seq_length, dim)
        h: filters of shape (seq_length, channels, dim)
        b: bias of shape (channels, dim)
    """
    h = rearrange(h, "l c d -> c d l")

    fft_size = seq_length * 2
    u_f = jnp.fft.rfft(x, n=fft_size)
    h_f = jnp.fft.rfft(h, n=fft_size)

    y = jnp.fft.irfft(h_f * x_f, n=fft_size, norm="forward")[
            ..., :seq_length]       # (c, d, l)
    y = y + x * b[..., None]        # (c, d, l)
    y = rearrange(y, "c d l -> l d c")
    return y
```

Pseudocode 1: Unstructured filters and convolutions.

```python
"""Context state collection for BST-SH variant."""

num_heads = 8       # (h)
num_states = 32     # (s)

# (SH): Single-Head
def SH_context_states(u):
    """Single-Head Context Collection."""
    h, b = get_filters_[unstruct/s4](channels=1)
```

```
    y_1 = multichannel_convolution(u, h, b)  # y_1: (l, d, 1)

    # lift to multiple heads
    y_h = dense(y_1)  # y_h: (l, d, h)

    context_states = jnp.split(
            y_h, seq_length // win_length, axis=0)
    return context_states # (l/w, w, d, h)
```

Pseudocode 2: Context state collection for BST-SH variants.

```
"""Context state collection for BST-MH variant."""

# (MH): Multi-Head
def MH_context_states(u):
    """Multi-Head Context Collection."""
    h, b = get_filters_[unstruct/s4](channels=num_heads)
    y_h = multichannel_convolution(u, h, b)  # y_h: (l, d, h)

    context_states = jnp.split(
            y_h, seq_length // win_length, axis=0)
    return context_states # (l/w, w, d, h)
```

Pseudocode 3: Context state collection for BST-MH variants.

```
"""Context state collection for BST-MF variant."""

# (MF): Multi-Filter
def MF_context_states(u):
    """Multi-Filter Context Collection."""
    h, b = get_filters_[unstruct/s4](channels=num_states)
    y_s = multichannel_convolution(u, h, b)  # y_s: (l, d, s)
    context_states = jnp.split(
            y_s, seq_length // win_length, axis=0)
    # context_states: (l/w, w, d, s)

    # collect the last context states
    context_states = context_states[:, -1, ...] # (l/w, d, s)
    context_states = rearrange(
            context_states, "lw d s -> lw s d")

    # shift context states corresponding to windows
    context_states = jnp.roll(context_states, 1, axis=1)

    # replace the initial window with trainable weights
    init_context = get_init_context(num_states) # (d, s)
    context_states[0] = init_context

    # lift to multiple heads
    context_states = dense(context_states)

    return context_states # (l/w, s, d, h)
```

Pseudocode 4: Context state collection for BST-MF variants.

```
"""Block-State Transformer Layer."""

# Block Transformers are non-recurrent and parallelizable.
block_transformer = jax.vmap(BRecT.nonrecurrent_cell)

def BST(u):
    """Block-State Transformer Layer."""
```

```
global MF # True if Multi-Filter, False otherwise (SH/MH)

# split inputs into windows (l/w, w, d)
u = jnp.split(u, seq_length // win_length, axis=0)

# collect context states from SSM outputs
context_states = [SH/MH/MF]_context_states(u)

# pass the contexts in place of recurrent states
y = block_transformer(
        token_embeddings=u,
        recurrent_state=context_states,
        use_cross_attn_causal_mask=not MF,
        use_cross_positional_emb=MF, # context IDs
)

return rearrange(y, "lw w d -> (lw w) d") # (l, d)
```

Pseudocode 5: Block-State Transformer Layer.

