# OpenReview forum: "Block-State Transformers"
_NeurIPS.cc/2023/Conference — NeurIPS 2023 poster_

### Official Review · Reviewer_4BCd · 2023-06-09

**Soundness:** 2 fair
**Presentation:** 3 good
**Contribution:** 3 good
**Rating:** 6
**Confidence:** 5

**Summary:**

The authors present a new long-range transformer architecture by incorporating SSMs. This novel model outperforms several established baselines, such as Transformer XL, Block Recurrent Transformer, and Sliding Window Transformer, in terms of cost-effectiveness trade-off for tasks involving long-document or code modeling.

**Strengths:**

1) Well-motivated. Long-range modeling is becoming increasingly important for LLM community.
2) Good results on language modeling (PG19, arXiv, Github)
3) The writing is clear and effectively conveys the ideas and findings.
4) The model design is intuitive and well-reasoned. The inclusion of both local full attention and linear components to handle long sequences is a sensible approach.

**Weaknesses:**

1) The scale is too small. For language modeling, based on the success of LLM, we always expect good scalability. This paper only conduct experiments up to 380M params, which are far from many emergent abilities threshold. When scaling up, many inductive bias would become useless[1].
2) More insightful experiments beyond language modeling are required. For instance, as the authors mentioned in the limitation section, what about the results on Long Range Arena?
3) Any case study about how this model captures long-range dependency? Why this model is indeed better? It seems that putting one efficient attention layer with linear or sub-linear complexity before self-attention should work similarly.
4) Some strong baselines like CoLT5[2] are missing.

[1] Scaling Laws vs Model Architectures: How does Inductive Bias Influence Scaling?
[2] CoLT5: Faster Long-Range Transformers with Conditional Computation

I enjoyed reading this idea but I believe the missed experiments above, especially (1), would be highly desirable. Without a set of experiments about scaling the model up, I cannot agree this paper is useful enough.


**Questions:**

1) Did the authors observe any instable training process? And again, would it become instable when scaling up?


**Limitations:**

See weakness

---

> ### Author Rebuttal · Authors · 2023-08-08
>
> Thank you for your valuable feedback. We have taken your comments and concerns into careful consideration and conducted additional experiments to address them. These experiments have been included in the **1-page PDF**, focusing on scaling aspects and performance in areas beyond language tasks, namely experiments that assess long-range modeling capabilities and scaling properties of BST. We would also like to draw the reviewer’s attention to the results in **Appendix C** which also provide insights into the scaling properties of our model.
>
> ---
>
> > "The scale is too small."
>
> > "This paper only conduct experiments up to 380M params."
>
> > "When scaling up, many inductive bias would become useless[1]."
>
> We acknowledge the growing significance of scalability in language models, and in response, we have incorporated additional scaling experiments in relation to the number of parameters in **Figure 1 of 1-page PDF**, where we scale BST from 80M to 1.3B parameters. We show a 0.1% relative perplexity improvement at 80M parameters, to a 3.8% relative perplexity improvement at 1.3B parameters over an equivalently large Block-Recurrent Transformer (BRT). These experiments demonstrate favorable scaling properties of our proposed model on perplexity. However in Table 1 of the paper all models have the same number of trainable parameters for a fair comparison.
>
> Ablation Studies in **Appendix C**, also suggest favorable scaling properties of our model. We provide experiments showing that performance uniformly improves as we add more contextualizing layers, i.e. BST layers, into the architecture. Moreover, an equally significant alternative scaling property pertains to the length of the sequence, and our model plays a direct role in addressing and contributing to this aspect, especially at inference time. See **Appendix B** in the supplementary material, on Evaluating Length Generalization capabilities.
>
> ---
>
> > "what about the results on Long Range Arena?"
>
> > "Any case study about how this model captures long-range dependency? Why this model is indeed better?"
>
> We thank the reviewer for suggesting experiments on Long-Range Arena (LRA), the results of which can be found in **Table 2 of the 1-page PDF**. Because the model’s ability to model long-range dependencies is not the only factor that influences perplexity, this experiment was necessary to directly demonstrate that our model captures long range dependencies better than the baseline (BRT). Further a fair comparison with Mega-chunk shows that we surpass on 4 out 6 tasks and on average the latest and strongest “chunked input” baseline on LRA.
>
> ---
>
> > "It seems that putting one efficient attention layer with linear or sub-linear complexity before self-attention should work similarly."
>
> As seen in Table 2 of the 1-page PDF, SSMs such as S4D, S4 and S5 are 30%-35% points higher on Long-Range Arena (LRA) compared Transformers with linear and sublinear attention (Linear Transformer, Performer, BigBird). It is therefore unlikely that we would gain any benefit from replacing the BST or SSM layers in hybrid models with linear or sub-linear attention layers.
> Further, transformer memories are fundamentally limited by their context length - $L$ (with $L^2$ complexity), whereas SSMs (like RNNs) can encode information indefinitely in their latent states. For this reason SSMs strongly outperform (linear) Transformers on long-range modeling tasks. Moreover, at inference time, Transformers completely lack the ability to generalize to unseen sequence lengths during training, unlike structured SSMs (See **Appendix B**).
>
> ---
>
> > "Some strong baselines like CoLT5[2] are missing."
>
> We were not aware of this baseline at the time of writing and submitting the paper. According to the arXiv publication date, this paper was made public on _March 17th_, which indicates that it is concurrent work that emerged during a similar timeframe as our research. This new work does not report perplexity on our targeted tasks nor LRA. We have cited CoLT5 and will nonetheless attempt to replicate and test CoLT5 on LRA and PG-19.
>
> ---
>
> > "Did the authors observe any unstable training process? And again, would it become instable when scaling up?"
>
> No, the model trains satably without the need of additional tricks.

---

> > ### Comment · Reviewer_4BCd · 2023-08-13
> >
> > Thanks for the additional experiments. The rebuttal solved most of my concerns well. Please do not forget to add these results to your paper. I do believe these experiments can greatly improve this paper. I have raised my score to 6. Good Luck!

---

### Official Review · Reviewer_rmDJ · 2023-07-04

**Soundness:** 2 fair
**Presentation:** 3 good
**Contribution:** 2 fair
**Rating:** 6
**Confidence:** 4

**Summary:**

This paper focuses on combining two efficient techniques for long-range modeling: state-space models (global contextualization) and block-recurrent transformers (local contextualization). In particular, they propose two different approaches, the first uses SSMs to output contexts for multiple heads (multi-head), and the second concatenates the last entries from the previous window to form a combined context state (multi-filter). Evaluation is performed on three language modeling datasets that outperform block-recurrent transformers in perplexity and is much faster when compared layer-wise.

**Strengths:**

S1. Exploring different ways to combine SSMs and block-recurrent models to improve efficiency is a compelling direction.  SSMs offer a parallelizable way to capture long-term information and avoid sequential computation in block-recurrent models. The results of this study should be of interest to researchers that study architectures that capture local and global information.

S2. The evaluation even though it focuses mainly on comparison with block-recurrent models and SSMs on three language modeling tasks, it is thoroughly described and well-executed.

**Weaknesses:**

W1. Even though the paper is mainly empirically driven, the delivery lacks a comprehensive and diverse set of evaluations to demonstrate the effectiveness and limitations of the method.

W2. Experiments are targetting language modeling on three tasks but there is no experiment that measures the long-range capabilities of the model. There are several long-context classification benchmarks that the authors can use in addition to language modeling: LRA [1], MuLD [2], and CAB [3].

W3. The method design makes specific assumptions about the hardware to be employed and bases its evaluation on it; e.g. efficiency comparisons are made per layer. It's not explored to what extent the benefit remains when comparing training time/speed vs quality for the whole model and evaluation is performed on typical accelerators. That reduces the practical impact in my view.

W4. A study regarding the behavior of the model when increasing the model size is missing. Scaling aspects are important to consider when making claims about outperforming transformers.

[1] https://arxiv.org/pdf/2011.04006.pdf
[2] https://arxiv.org/pdf/2202.07362.pdf
[3] https://arxiv.org/pdf/2210.07661.pdf


**Questions:**

Q1: Evaluating methods that model long context typically involves more tasks than language modeling. What is the level of confidence that proposed models perform to other tasks such as long context classification and text generation?

Q2:  It would be interesting to measure the time it takes for every method to converge even within a fixed time budget. Have all methods converged in the fixed training budget in Table 1 and do they have any differences worth discussing?



**Limitations:**

Discussion about the limitations of the proposed method would be useful, I'd suggest talking about scaling behavior, performance on conventional accelerators, and generalizability to long-context tasks.

---

> ### Author Rebuttal · Authors · 2023-08-08
>
> Thank you for your thoughtful feedback. In response to your comments and concerns, we have conducted additional experiments, which we have included in the **1-page PDF**. These experiments address the points you have raised and also explore areas that you alluded to in your review, namely scaling aspects and performance on Long-Range Arena (LRA), outside the language modeling domain.
>
> ---
>
> #### Re W1 and W2: Extending to areas other than language/perplexity
>
> As we acknowledged in the limitations section (Appendix D) of our submission, we recognize the significance of conducting experiments to assess the long-range capabilities of our proposed model. These new experiments are designed to test if the improved performance on language modeling can be attributed to our model’s ability to infer long-range dependencies. To test this hypothesis rigorously, we have performed comparisons between variants of our proposed model, BST, and its recurrent predecessor, BRT, as well as several other baselines that utilize SSMs. We use the Long-Range Arena (LRA) benchmark as the primary testbed. Kindly refer to **Table 1 of the 1-page PDF**. The results demonstrate that, in line with our expectations, SSMs enhance long-range modeling capabilities compared to BRT.
>
> ---
>
> #### Re W3: Efficiency of our model
>
> In the interest of convenience while maintaining an exploratory approach and ensuring a fair comparison to former baselines on the dataset, we intentionally chose not to utilize model parallelism, despite the fact that our model is well-suited for it. Unlike BRT, whose recurrent layers cannot be parallelized, our model’s per-layer speedups (Figure 3) will trivially translate to full model speedups provided that there is sufficient compute resources to support parallelism. **Appendix C, Table 3** shows that replacing transformer layers with BST layers monotonically improves perplexity performance. This evidence, combined with the per-layer speedup results, demonstrate the performance and speedup potential of our full model at scale.
>
> ---
>
> #### Re W4: Scaling capabilities
>
> Please see the additional scaling experiments in the **1-Page PDF, Figure 1**. Scaling experiments relate parameter count (80M to 1.3B) to the perplexity performance. We show a 0.1% relative perplexity improvement at 80M parameters to a 3.8% relative perplexity improvement at 1.3B parameters over an equivalently large Block-Recurrent Transformer (BRT) model. Additionally, ablation studies in **Appendix C Tables 2-4** relate the placement, capacity, and the number of the BST layers to performance.
>
> ---
>
> #### Re Q1: LRA
> > "What is the level of confidence that proposed models perform to other tasks such as long context classification and text generation?"
>
> We have benchmarked BST and BRT on Long-Range Arena in **Table 2 of the 1-page PDF**. Results show that BST is indeed better at capturing long-range relations, which aligns with previous results (e.g. S4 vs Transformers) and our initial motivating intuition.
>
> ---
>
> #### Re Q2: Training/Convergence Time
> > "Have all methods converged in the fixed training budget in Table 1 and do they have any differences worth discussing?"
>
> This is an interesting question. Our experiments show that BRT and BST both had room to continue improving and our validation set perplexity continued to diminish on all datasets. While our experimental setups were designed to align with prior works (GSS & BRT), this is a reasonable question and we are well positioned to include these additional experiments in the camera-ready version of the paper.

---

> > ### Comment · Reviewer_rmDJ · 2023-08-16
> >
> > Thank you for the rebuttal. The replies and additional experiments address my main concerns. To reflect this, I increased my score.

---

### Official Review · Reviewer_XDNb · 2023-07-06

**Soundness:** 3 good
**Presentation:** 3 good
**Contribution:** 3 good
**Rating:** 6
**Confidence:** 4

**Summary:**

State space models (SSMs) perform well on modeling long-range dependencies with good efficiency scaling, but on language modeling, transformers still outperforms SSMs. This paper tries to combine the best of both worlds and proposes a hybrid model, Block-State Transformer, which combine SSMs’ capacity on long range modeling and Transformer’s ability on modeling local context. The input sequence is split to multiple smaller segments. For each segment, transformer layer will do a self attention on this token embeddings and cross attention to the output of SSMs. Their experiments show that on language modeling, their approach achieve reasonable speedup with comparable performance to Transformers.

**Strengths:**

1. The proposed combination of SSMs and transformers allow the model to exploit advantages of two powerful methods while avoiding their drawbacks.
2. The SSMs used in the proposed method can be swapped to different SSMs making it possible to enjoy the advancement on SSMs field.
3. The proposed method give similar performance on language modeling compared to Transformers.


**Weaknesses:**

1. There was already existing work on SSMs that achieve similar performance on language modeling (https://arxiv.org/abs/2212.14052) compared to Transformers. The authors should include a discussion and comparison to the relevant work.
2. The evaluations are performed on language modeling for 4096 length sequences. On this setting, there are a lot of strong transformer baselines with efficient self-attention designs. It would be good if the authors can provide an empirical comparison with these baselines.
3. Perplexity is only one indicator of how the language models performance. To get more precise understanding of performance, it would be good to include a comparison on downstream tasks.
4. Code is not available.


**Questions:**

The main concerns are listed above. There are two more questions:
1. On line 235, the 6,966,499 English language words seems to be a typo according to dataset statistics on https://github.com/deepmind/pg19.
2. What are the latency of one forward step and what about memory consumption?

**Limitations:**

Yes.

---

> ### Author Rebuttal · Authors · 2023-08-08
>
> Thank you for your generally positive review. We have taken your valuable feedback into account to improve our current version of the paper. A more comprehensive review of related works, including H3, will be provided. Additionally, we have included a number of experiments in the **1-page PDF**, to further assess long-range modeling capabilities (LRA) and scaling properties of our model. Moreover, we have expanded our baselines on language to also include Hyena, a more recent and general framework that subsumes H3 and GSS.
>
> Our additional experiments show:
> - Large improvements on PG-19 compared to Hyena and Hybrid H3. Specifically, integrating parameterized convolutions and attention in `BST:SH:unstruct` allows our model to **surpass Hyena by 1.4 perplexity points**. See **Table 1**.
> - ***+30%-35% point improvement on Long-Range Arena (LRA)*** over recent and strong Transformer baselines. See **Table 2**.
>
> ___
>
> > W1. "There was already existing work on SSMs that achieve similar performance on language modeling (https://arxiv.org/abs/2212.14052) compared to Transformers."
>
> We have added Hybrid-H3 and Hyena as additional baselines in the **Table 1 of the 1-Page PDF**.
>
> Hyena Hierarchy (https://arxiv.org/pdf/2302.10866.pdf) is a language model that draws inspiration from GSS and H3, and captures both of these methods within a more generic framework. In all these approaches, attention and retrieval are conceptually simulated by element-wise multiplication of a sequence of tokens, and its corresponding contextualized (via SSM or parameterized convolutional kernels) counterpart. Hyena, outperforms both GSS and H3 on tasks such as associative recall and in language modeling, particularly on the WikiText103 dataset.
>
> We have trained both BST and BRT models under a fixed parameter setting using the same tokenizer (GPT2) and vocab size, for a fair comparison. BST achieves SoTA performance on this dataset. Results can be found in **Table 1 of the 1-Page PDF**. We hope this addresses your concern regarding benchmarking SSM-inspired language models.
>
> ---
>
> > W2. "The evaluations are performed on language modeling for 4096 length sequences. On this setting, there are a lot of strong transformer baselines with efficient self-attention designs."
>
> To the best of our knowledge, GSS and BST were the state-of-the-art on the datasets we use in our work, outperforming other efficient Transformers, e.g. linear implementation etc. More importantly, other Transformer-based architectures do not generalize to sequence lengths not seen during training, whereas our method does (See **Figure 4 in Appendix B**). Even with relative positional embeddings, Transformers cannot reliably go beyond 3x the trained sequence length (see https://arxiv.org/abs/2305.19466), let alone 16x (at a sequence length of 65K for example). That being said, if the reviewer can point us to specific works, we are well-positioned to include additional baselines in the camera-ready version of the paper. Further, as seen in **Table 2 of the 1-Page PDF**, our method greatly outperforms very recent and efficient Transformers such as Linear Transformer, Reformer, Performer and BigBird.
>
> ---
>
> > W3. "Perplexity is only one indicator of how the language models performance."
>
> Although there is no unanimous consensus, most practitioners in the field generally agree that the performance on downstream tasks seems to be well correlated with perplexity for LLMs (https://arxiv.org/pdf/2210.14199.pdf, https://aclanthology.org/2021.emnlp-main.478.pdf). Like other works in this field, we focus on developing decoder-only models that achieve lower perplexity. Nevertheless, we acknowledge that this is an important step in developing large language models and believe BST can also serve as a powerful encoder model which may be evaluated on downstream tasks. Because BST uses off-the-shelf transformer models augmented with SSM states as context, our approach can be easily adapted into existing LLM codebases. While no one has evaluated SSM or hybrid-SSM pretraining performance on downstream tasks yet, we look forward to doing that with BST in a follow-up project.
>
> ---
>
> > W4. "Code is not available."
>
> As part of our ongoing work, we are working closely with the maintainers of the Block-Recurrent Transformer codebase to integrate our implementation of BST into the repository (https://github.com/google-research/meliad). In the meantime, we have provided the JAX pseudo-code for all of our variants in **Appendix E** in the supplementary materials.
>
> ---
>
> > Q1. "On line 235, the 6,966,499 English language words seems to be a typo..."
>
> Thank you for pointing this out. We have fixed this error in the latest version.
>
> ---
>
> > Q2. "What are the latency of one forward step and what about memory consumption?"
>
> The computational and space complexity of a BST layer consists of that of the transformer blocks and the SSM sublayer. We discuss this topic in more detail in the Efficiency section of the paper. In the left-side plot of Figure 3 in the paper, the y-axis represents the latency for a forward pass of our model when executed on an NVIDIA V100 GPU.
>
> We assume that your question refers to the auto-regressive “forward step” at test time. The exact forward step latency may vary slightly depending on the BST variant used. In Single-Head and Multi-Head variants, every token generation step is immediately followed by feeding that token back into the SSM and adding it to the context in $\mathcal{O}(1)$ operations – via the RNN view of SSMs. On the other hand, when the Multi-Filter variant is used, contextualizing can be postponed to when all the tokens corresponding to the current block are generated. The new block of tokens are then added to the context in one go, using the recurrent view of SSMs in $\mathcal{O}(W)$ operations. The token generation done within the Transformer is similar to BRT, $\mathcal{O}(W)$, since every token attends to $W$ previous tokens.

---

> > ### Comment · Reviewer_XDNb · 2023-08-19
> >
> > Thank you for the efforts on rebuttal. The response addressed my concerns. I will maintain my score.

---

> > > ### Author Response · Authors · 2023-08-19
> > > **Thank you for reading our response**
> > >
> > > We really appreciate your reply.
> > > As you mentioned, the response addresses your concerns and this includes:
> > > - Adding more explanations on prior work such as H3, and Hyena.
> > > - Showing that BST outperforms other efficient transformers on Long Range Arena (see *Table 2 of the 1-Page PDF*).
> > > - Detailing memory consumption at inference in $\mathcal{O}(1)$ for single head BST.
> > >
> > > We have added additional experiments as well that we hope will make our submission even stronger.
> > >
> > >
> > > If you feel that your concerns have been adequately addressed, is there any specific feedback that we should discuss to improve our submission and increase our score?

---

### Official Review · Reviewer_9Xt3 · 2023-07-06

**Soundness:** 2 fair
**Presentation:** 2 fair
**Contribution:** 2 fair
**Rating:** 5
**Confidence:** 5

**Summary:**

This paper proposes block-state transformers, a method to combine state space models with transformers for language modeling. The paper evaluates block-state transformers on PG19 and arxiv math and finds promising results.

**Strengths:**

Combining state space models and Transformers is an interesting idea worth exploring. The presentation of the paper is clear. The explanation of state space models, which can be quite complex, is very clear. The evaluation hits the right notes in terms of the major questions to ask.

**Weaknesses:**

The evaluation is missing many recent works combining state space models and attention in various ways. The claim that SSMs do not match Transformers on language has not been true for a while. Most of these methods were released significantly before the NeurIPS deadline and are critical to compare against for evaluation.

* Mega [1] combines attention and state space models.
* BiGS [2] is a new SSM-based architecture that matches Transformers in language.
* H3 [3] combines SSMs and attention in alternate layers.
* Hyena [4] removes attention completely and replaces it with a convolution-based layer (similar to an SSM).

Confusingly, many of these works are cited in the paper - and ideas from the papers are used extensively in the methods proposed (e.g., "BST:{SH,MF}" uses the structure from H3 and Hyena without comparing against those architectures as baselines). Using the ideas from these papers without comparing against them makes it difficult to understand how this method compares against previous methods and where the performance improvement comes from.

Performance is also hard to evaluate compared to standard models such as Transformers (GPT-Neo [5], Pythia suite [6]). TransformerXL is an older model that is not trained as well as modern Transformer-based LLMs.

[1] https://arxiv.org/abs/2209.10655
[2] https://arxiv.org/abs/2212.10544
[3] https://arxiv.org/abs/2212.14052
[4] https://arxiv.org/abs/2302.10866
[5] https://github.com/EleutherAI/gpt-neo
[6] https://github.com/EleutherAI/pythia

**Questions:**

How does BST compare to the architectures listed in the weaknesses section? It is important to compare against the original architectures that inspired components of BST, as well as modern standard Transformers.

**Limitations:**

The paper would be stronger with more discussion of limitations.

---

> ### Author Rebuttal · Authors · 2023-08-08
>
> Thank you for your review. We hope that we have addressed most of your concerns with the additional experiments and comparisons in the **1-page PDF**.
>
> Our new experiments show:
>
> 1. Large improvements on PG-19 compared to Hyena and Hybrid-H3. Specifically, integrating Hyena and attention in `BST:SH:unstruct` allows our model to surpass standalone Hyena ***by ~1.4 perplexity points***. See **Table 1 in 1-page PDF**.
>
> 2. We show that we surpass all Transformer variants that you have mentioned and other methods that chunk inputs (allowing similar speed-ups to our BST) such as Mega-chunk. Specifically, on Long-Range Arena (LRA), `BST:SH:S4` performs better than:
>     - S4 on 5 out of 6 tasks and on average by 0.9% points
>     - Mega-chunk on 4 out of 6 tasks and on average by 1.3% points
>
> Furthermore, we respond to your specific questions and comments below.
>
> ---
>
> > “Mega [1] combines attention and state space models.”
>
> Thanks for bringing Mega to our attention, we will review and cite it in our related works section.
>
> We compare our model to Mega on Long-Range Arena (LRA), where we see that our model achieves comparable performance to Mega/Mega-chunk. However on language modeling, according to Table 1 in [1], **Mega** (252M) achieves _18.07_ perplexity on WikiText103 that is roughly the same performance, _18.50_ perplexity, achieved by **Hybrid-H3** [3] (125M, half the size of Mega). ***`BST:SH:unstruct` outperforms Hybrid-H3 by 3.0 perplexity points*** on the PG-19 long text language modeling benchmark, see **Table 1 in the 1-page PDF**. Therefore, although we were unable to directly compare our model against Mega on Language Modeling in the rebuttal phase, we have reason to believe that our model will outperform Mega when using the same number of parameters.
>
> ---
>
> > "BiGS [2] is a new SSM-based architecture that matches Transformers in language."
>
> Regarding BiGS, we consider this work to be less relevant compared to the other papers mentioned. The reason is that BiGS is designed as a bidirectional model, making it more suitable for serving as an encoder or for natural language understanding tasks, rather than auto-regressive language modeling or language generation tasks. In contrast, our focus is on decoder-only language models, which have different requirements and objectives. Further, BiGS is not evaluated on Long-Range Arena which makes it difficult to compare against.
>
> ---
>
> > "H3 [3] combines SSMs and attention in alternate layers."
> > "Hyena [4] removes attention completely and replaces it with a convolution-based layer (similar to an SSM)."
>
> We have compared BST to H3-Hybrid and Hyena on PG-19, results can be found in **Table 1 of the 1-page PDF**. We have trained both BST and BRT models under a fixed parameter setting using the same tokenizer (GPT2) and vocab size, for a fair comparison. BST remains to be state-of-the-art at this scale. Specifically, integrating Hyena and attention in `BST:SH:unstruct` allows our model to surpass standalone Hyena by ~2 perplexity points.
>
> ---
>
> > "Confusingly, many of these works are cited in the paper - and ideas from the papers are used extensively in the methods proposed (e.g., "BST:{SH,MF}" uses the structure from H3 and Hyena without comparing against those architectures as baselines)."
>
> Please see **Table 1 in 1-page PDF** for a direct comparison against H3 and Hyena where our model outperforms both under a fixed parameter count setting.
>
> ---
>
> > "Performance is also hard to evaluate compared to standard models such as Transformers (GPT-Neo [5], Pythia suite [6])."
>
> As we opted to implement our models using JAX, we were unable to utilize other open-source PyTorch-based codebases like GPT-Neo and Pythia by the rebuttal deadline. However, we should have such comparisons before the camera-ready deadline. It is crucial to mention that any enhancements made to the attention layer are independent/orthogonal to our model which focuses on enabling off-the-shelf transformer-attention layers to model long-range dependencies.
>
> Furthermore, we want to note that since GPT-Neo and Pythia employ pipeline parallelism, we anticipate similar speed improvements when applying SSMs since they can be parallelized in a similar way. As part of our ongoing work, we are actively developing a PyTorch implementation for BST and BRT. These implementations will enable us to reliably compare performance with other PyTorch models and explore their potential benefits in conjunction with our proposed method. Please note that we do evaluate performance against linear transformer, reformer, performer and BigBird in **Table 2 of 1-pagePDF**.

---

> > ### Comment · Reviewer_9Xt3 · 2023-08-11
> >
> > Thank you for the extensive rebuttal and extra experiments. I will be raising my score to a 5.
> >
> > For the experiments, one way they can be improved: Hyena and H3-Hybrid were trained on the Pile as their primary evaluation, and report 5B, 10B, and 15B-token experiments. It would be helpful to compare against those to ensure a fair comparison.

---

> > > ### Author Response · Authors · 2023-08-17
> > >
> > > Thank you for responding to our rebuttal and increasing your score.
> > >
> > > Hyena and H3 have indeed different results on the PILE which are not directly comparable since the experimental set-up was different in each paper. For example, Hyena experiments are up to 15B tokens and H3 experiments are with 400B tokens only. Therefore, we cannot directly compare Hybrid-H3 and Hyena results on PILE from their papers.
> > >
> > > However, Hyena and Hybrid-H3 experiments on PG-19 are equivalent and we have demonstrated superior BST performance (see Table 1 in 1-page PDF). Further, with Wikitext103, we can also fairly compare Hyena and Hybrid-H3. From Table 4.3 of the Hyena paper (https://arxiv.org/abs/2302.10866), we see that the perplexity of Transformer and Hyena are on-par and that Hybrid-H3 improves over the Transformer baseline by only 0.5%.
> > >
> > > However, within our experiments (see Table 1 of our paper), we find generally that: BRT improves over the Transformer baseline by 2.1% (average over PG-19, GitHub and arXiv).
> > >
> > > We find the performance improvement to be much larger between BST and the Transformer baseline compared to either Hyena and Hybrid-H3.
> > >
> > > Does this answer your question, and resolve your remaining concerns?

---

### Official Review · Reviewer_V1Sd · 2023-07-08

**Soundness:** 3 good
**Presentation:** 2 fair
**Contribution:** 3 good
**Rating:** 6
**Confidence:** 4

**Summary:**

The authors present a novel architectural framework called the Block-State Transformer (BST), which integrates State Space models and Block-Recurrent Transformers to create a competitive autoregressive language model capable of effectively processing lengthy sequences. The input sequence is passed through a State Space model like S4, and the output of which is later in Block-recurrent Transformer as a replacement of the recurrent state vectors. To obtain the final output, the input embeddings are divided into fixed-sized windows and processed in parallel by a series of Block-Recurrent Transformers. Due to the usage of the S4 output as recurrent state vectors within the Block-Recurrent Transformers, the absence of recurrence allows for parallel computation. The authors propose three distinct integration modes, which differ in terms of how the S4 output is integrated within the recurrent state vectors. To evaluate the performance of BST, the authors compare it against four baseline models: Transformer-XL, Slide, Block-Recurrent Transformer, and a hybrid Gated State Space model. The comparison is conducted across three diverse datasets, namely PG19, arXiv Math, and GitHub. BST demonstrates slight perplexity improvements in the PG19 and GitHub datasets. Additionally, the authors present ablation studies on various parameters, including SSM layer placement, the number of SMM layers, and the state dimensionality of SMM.

**Strengths:**

+ **Strong Presented Results**: Authors presented results on competitive benchmarks across reasonable prior baselines such as Transformer-XL, Slide, Block-Recurrent Transformers etc. and do outperform them across several tasks.

+ **Computational efficiency**: The proposed method is able to provide a huge improvement in terms of computational efficiency over models like Block-Recurrent Transformers by parallelization.

+ **Interesting combination of prior ideas**: By using the parallelizable nature of the SSMs the authors were able to introduce parallelization to Block-Recurrent Transformers thereby achieving computation efficiency.

**Weaknesses:**

- **Incomplete Related works**: The authors' treatment of related works, particularly in the context of models combining Transformers and S4 appears to be lacking. It would have been beneficial for the authors to provide a more comprehensive discussion on existing models that incorporate both Transformers and S4. This would have allowed for a deeper exploration of the advancements and limitations of these models, highlighting the unique contributions of the proposed Block-State Transformer (BST). In addition to that, there could be more details on SSM development in related works since S4, S5, and S4D were mentioned in the paper later.

- **Missing Preliminaries on S4/S5** : A more detailed description of the S4 models within the method section would have been beneficial, particularly regarding the computation of the kernel since the complexity of the S4 model is the major part of the BST model complexity. Specifically, the computation of the kernel is not trivial if you want to keep overall L log(L) complexity and it relies on the form of the A and B matrices. A more comprehensive exposition regarding the computational aspects of the S4 models is deemed necessary for a thorough understanding of the subject matter.

- **Additional benchmarking** : As the authors themselves admit in the limitations section, there are further results required, especially on well benchmarked domains such as the Long Range Arena and also other long-term datasets to provide convincing evidence of BST performance.

**Questions:**

Broadly, understanding BST capabilities in other settings would help enrich the paper results. See weaknesses.

---

> ### Author Rebuttal · Authors · 2023-08-08
>
> We appreciate your valuable and constructive comments. We agree that Block-State Transformer is a novel architecture that shows strong results and computational efficiency. We think that we have only scratched the surface of possibilities with this interesting combination of ideas. We have conducted more experiments that can be found in the **1-page PDF** (see general comments to access PDF), some of which are related to your review, namely benchmarking BRT and BST on Long-Range Arena. Additional experiments also include scaling results and more baselines on language tasks. We believe that these supplementary results address most of your  feedback on proving that we can harness the strengths of S4 and Transformer in long-range classification and language modeling tasks.
>
> ___
>
> #### Extending Related works
> We have indeed included a more comprehensive review of related work in the latest version of our paper, while also emphasizing the key differences. Our discussion will include S4 kernel computation, GSS-Hybrid, Hybrid-H3, and Mega. We have already outlined some of the core contributions in [response to Reviewer 9Xt3](https://openreview.net/forum?id=XRTxIBs2eu&noteId=wjrwjedtwd), and we invite you to review them for further insights.
>
> ---
>
> #### Covering Preliminaries
>
> We have incorporated your feedback by expanding the discussion in the efficiency section in the latest version of the paper. To summarize, the efficiency of our models is largely afforded by standard SSM implementations, which are due to:
>
> * Imposing Diagonal Plus Low-Rank (DPLR) structure on the $A$, $B$, and $C$ matrices, rematerializing the convolutional kernel can be carried out in tandem with the convolution operation with $\mathcal{O}(L \log L)$ complexity, as described in S4.
> * By replacing RNNs (BRT) with SSMs (BST), BST transformer blocks can run in parallel instead of sequentially as in BRT.
>
> Further, by demonstrating two vastly different SSM as means of contextualization, we show that our proposed architecture/framework remains agnostic to the specific type of SSM used, making the advancement of SSMs orthogonal to our work. That being said, and as highlighted in your review, since the computational efficiency of BST layers _can be_ dictated by the SSM, it is worth discussing them in more detail.
>
> ---
>
> #### Additional benchmarking
>
> Please see **Table 2 in the 1-page PDF** for additional experiments on Long-Range Arena (LRA).
>
> Our motivating hypothesis was that replacing recurrence with a more efficient yet powerful component for capturing long-range dependencies would improve performance on language modeling tasks. As we openly acknowledged in the limitations section, we recognized the need for more concrete evidence to support this hypothesis. Consequently, in response to your request, we conducted additional experiments on Long-Range Arena (LRA) (please see **Table 2 of 1-page PDF**), demonstrating that BST indeed surpasses BRT (and other methods that chunk inputs such as Mega-chunk) in capturing long-range dependencies.
>
> ---
>
> Thank you for reading our response!

---

### Author Rebuttal · Authors · 2023-08-09

We would like to thank all the reviewers for their insightful comments. We believe that we addressed the vast majority of reviewer’s concerns by conducting additional experiments, found in the attached **1-page PDF attached to this message**, and responding to reviewer’s individual questions. Our additional experiments include:


#### **Scaling Properties of Block-State Transformer (BST)** [_Figure 1_]

- Showing the competitive scaling properties of Block-State Transformer (BST), going from a 0.1% relative perplexity improvement at 80M parameters to a ***3.8% relative perplexity improvement at 1.3B parameters*** over an equivalently large Block-Recurrent Transformer (BRT).


#### **Comparisons to Hybrid-H3 and Hyena** [_Table 1_]

- Demonstrating that our model achieves ***superior perplexity*** compared to Hyna & Hybrid:H3, respectively by 1.4 and 3.0 points, under standard evaluation conditions (fixed parameter count setting).


#### **Comparisons to Transformer variants and Mega on Long-Range Arena (LRA)**  [_Table 2_]

- Achieving a substantial lead over the Block-Recurrent Transformer (BRT). Further, BST outperforms Transformer variants **by 30%-35% points on average**. Further, in a fair comparison with other methods that chunk input sequences, we see that BST outperforms Mega-chunk **on 4 out of 6 tasks** and by 1.3% point on average.

Our strong results on Long-Range Arena (LRA) demonstrate that BST significantly outperforms BRT on LRA, providing further evidence that the gains on language modeling tasks can indeed be attributed to our model’s ability to capture long-range dependencies more effectively while improving computational efficiency.

---

We have included the responses to the reviewers in their rebuttal sections individually. We would like to thank all reviewers again for their time and attention.

---

Find the PDF containing additional results below.

---

### Decision · Program_Chairs · 2023-09-21

**Decision:**

Accept (poster)

**Comment:**

This paper introduces the Block-State Transformer (BST), a novel hybrid layer that combines state space models (SSMs) for long-range contextualization and Block Transformer sublayer for short-term sequence representation. The authors propose three parallelizable variants for integrating SSMs and block-wise attention. The reviewers are overall positive towards the contribution of this paper. I would recommend to accept this paper.